# Aligning Forest and Trees in Images and Long Captions for Cross-Domain Grounding

## Abstract

Large vision-language models such as CLIP align images and captions as wholes but falter on long, detailed descriptions. Fine-grained understanding demands capture of hierarchical semantics, *seeing both forest and trees*, within and across domains. Yet syntactic and semantic structures seldom mirror visual organization, and vision alone tends to create spurious fragments unless text anchors and unifies.

We propose *F-CAST*, a hierarchical image-text representation learning framework that discovers aligned *spatially oriented text* and *visual hierarchies* directly from image and long-caption corpora, without region-sentence labels. It uses a CAST visual encoder for fine-to-coarse scene parsing and a hierarchical transformer text encoder that first encodes each sentence then fuses them into a whole-caption representation. A *two-level alignment loss*, extending FLAIR, aligns whole images with whole texts while biasing image-sentence matches so coarse concepts emerge from fine-grained evidence rather than ignoring it.

Trained on 30M image–text pairs, F-CAST delivers strong scaling and sets state-of-the-art performance on six long-text benchmarks. Experiments show that hierarchical alignment of vision and language enables F-CAST to discover fine-grained, visually grounded text understanding without supervision.

## 1 Introduction

A *red bus* is never just a *red bus* (Figure 1): *a modern red double-decker, labeled route 38, front angled toward the camera, pedestrians nearby, a hatchback passing*. Such long text can be both a blessing and a trap: every extra clause adds information but also noise. Given four photos of London buses, a model that merely matches whole captions to whole images may seize on the wrong clue (Radford et al., 2021; Zhang et al., 2024; Wu et al., 2024). To get it right, a system must see *forest and trees* both *within* and *across* two domains (Biederman, 1987; Maurer et al., 2002).

---

This is an image of **a modern red double-decker bus**, specifically labeled route 38, on a city street. The front of the bus is angled towards the camera, showing its distinctive curved design and large glass windscreen. The bus has a black trim around the base and wheel arches, and it features the typical London transport logo. Pedestrians are on the pavement to the left, with one person stepping onto the bus. Behind the double-decker, another red bus is visible. A red hatchback car drives by on the road, and behind it is a stone building with arched windows and a flagpole on its roof. The sky is partly cloudy.

---

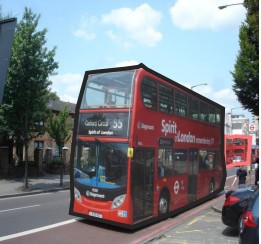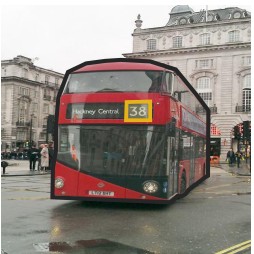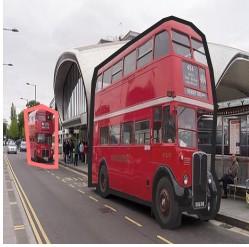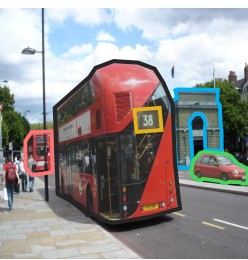

Figure 1: **Long-text image retrieval with hierarchical grounding.** How to find the image that matches a long caption? It requires locating the image region that grounds each descriptive phrase. This illustrates that long-text image understanding hinges on aligning the *text hierarchy* (whole caption and its sub-phrases) with the *image hierarchy* (whole scene and its constituent regions).

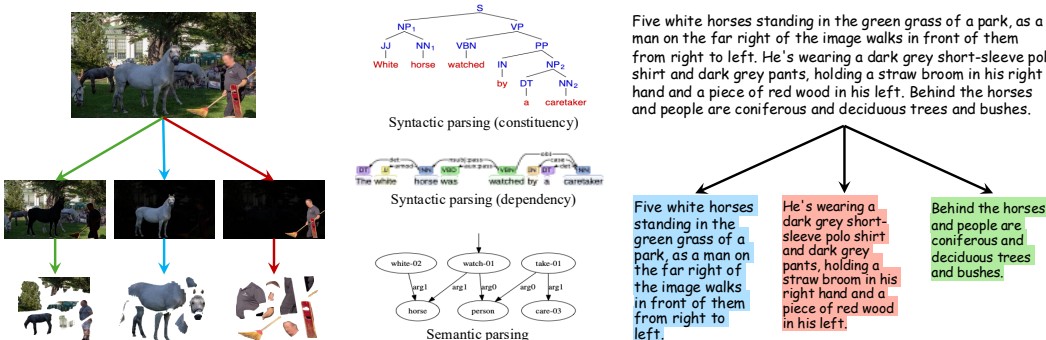

Figure 2: **Fine-grained image-text understanding requires aligned hierarchical parsing in both domains.** Parse trees capture grammar and semantic graphs capture events (center). The spatially composed visual hierarchy (left) must be matched by a spatially oriented text hierarchy (right).

Visual parsing needs to group *image patches* into *objects*, and *objects* into a *scene*, while tracking relations like *pedestrians stepping aboard* and *traffic moving past* (Fukushima, 1980; Kuzovkin et al., 2018). Language can guide this process, but not in its standard *syntactic* or *semantic* guises (Figure 2). *Parse trees* describe grammar (Shi et al., 2019; Drozdov et al., 2019). *Semantic graphs* capture events and relationships rather than the geometric assembly of *buses*, *cars*, and *buildings* (Johnson et al., 2015; Zellers et al., 2018; Yang et al., 2018). **We need a text hierarchy that mirrors how scenes are built, where each *sentence* can point to a *visual component*** (Figure 3).

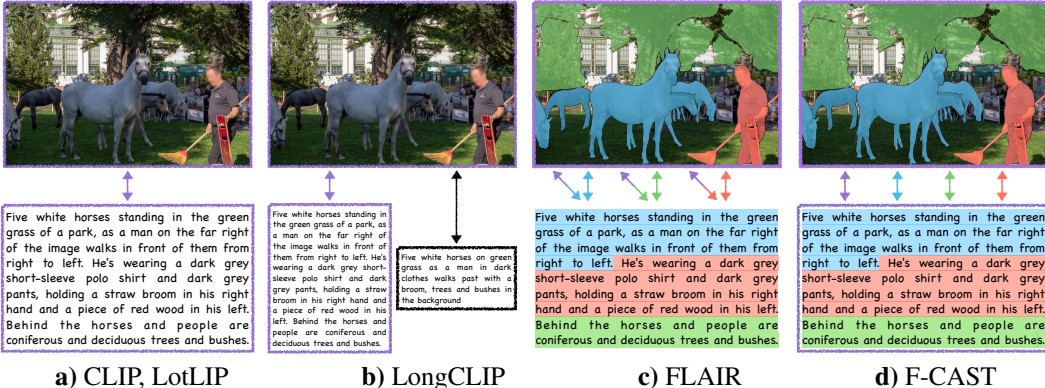

**a)** CLIP, LotLIP          **b)** LongCLIP          **c)** FLAIR          **d)** F-CAST

Figure 3: **Cross-domain alignment at whole and part levels.** Many methods aim to match text and image features over the same-colored regions. They focus on full-image understanding (**a,b**) or both whole and parts (**c**), confounding semantic and spatial hierarchies. Our method aligns semantic hierarchy with spatial hierarchy across both domains (**d**).

We propose F-CAST based on three design insights. **1) Spatially oriented long-caption data.** Many long captions consist of sentences that each describe a distinct object or scene component. We exploit this structure to form a spatial text hierarchy where each sentence can match an image region counterpart, without supervision. **2) Consistent within-domain hierarchy.** Images and texts are decomposed so that coarse levels are explicitly built from finer ones, ensuring that holistic understanding grows from detailed evidence. We design a *hierarchical transformer* text encoder inspired by HAT (Chalkidis et al., 2022): Each sentence is first encoded in parallel, and its embeddings are then combined in a second stage to form the representation of the full description. We adopt CAST (Ke et al., 2022) as the visual backbone to capture this structure through concurrent fine-to-coarse segmentation, so a *red double-decker bus labeled route 38* is never visually collapsed into merely *a bus*. **3) Two-level cross-domain alignment loss.** F-CAST discovers aligned parts and wholes in image and text domains through a hierarchical alignment objective. We adopt FLAIR (Xiao et al., 2025) and extend its text-conditioned image representation so that mid-level visual features align with part-text embeddings for localized grounding, and top-level visual features align with whole-text embeddings to capture global scene semantics.

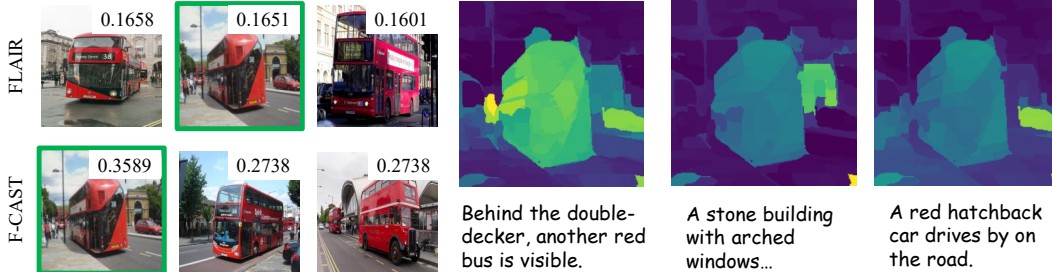

Figure 4: **Our F-CAST demonstrates fine-grained visually grounded long-text understanding**. **Left)** Top 3 text-to-image retrievals show that FLAIR wavers among look-alikes, while F-CAST picks the clear winner. **Right)** Attention visualization for each sub-caption shows that our gain stems from visually grounding sub-sentences during recognition, discovered entirely without supervision.

Trained on 30M image-text pairs, F-CAST scales well and sets new records on six long-text retrieval benchmarks. Ablation studies show that its hierarchical design matters: Modeling visual and linguistic structure sharpens recognition, and part-to-whole alignment delivers the biggest gain. Experiments confirm that aligning vision and language through consistent hierarchies lets F-CAST discover fine-grained, visually grounded text understanding without supervision (Figure 4).

## 2 RELATED WORKS

**CLIP with long captions** has recently emerged as a promising direction, enabled by synthetic long captions generated by MLLMs (Zheng et al., 2024; Chen et al., 2024). These longer captions provide richer supervision than raw web annotations. To adapt longer input, several studies (Zhang et al., 2024; Najdenkoska et al., 2024; Choi et al., 2025; Asokan et al., 2025) extend CLIP's capacity by enlarging the positional encoding from 77 to 248 tokens, while LoTLIP (Wu et al., 2024) explores this from scratch. However, these methods struggle to capture both the local details in sub-captions and the holistic semantics of the full caption. To address this limitation, we propose a *hierarchical transformer* text encoder that hierarchically captures both local and holistic semantics.

**Fine-grained vision–language understanding** has been explored to improve models across many downstream tasks (Antol et al., 2015; Vinyals et al., 2015; Pak et al., 2024; Wang et al., 2025). Such approaches typically ground local elements, such as noun phrases, with dense annotations (e.g., bounding boxes). For example, GLIP (Li et al., 2022) aligns phrases with regions using large-scale human-labeled data, while GOAL (Choi et al., 2025) employs SAM to obtain regions of interest (Kirillov et al., 2023) and aligns them with sub-captions through a pretrained CLIP. More recently, FLAIR introduces text-conditioned attention pooling to localize sentences on visual patches, enabling fine-grained representations without dense labels. Building on FLAIR's attention pooling, our F-CAST also learns such implicit grounding but further integrates fine-grained regions into holistic semantics—capabilities not explored in prior vision–language pretraining.

**Leveraging hierarchy in vision-language model training** has been scarcely studied, due to matching difficulty. HiCLIP (Geng et al., 2023) employs a hierarchical architecture for vision and text, uncovering part-to-whole hierarchy and thereby improving fine-grained recognition. However, Hi-CLIP's recognition objective is applied only at the top layer, limiting its ability to fully exploit the hierarchy. Both HiVLP (Chen et al., 2022) and HieCLIP (Hua et al., 2025) propose multi-level alignment between image and text, but their representations lack an explicit part–whole structure, resulting in suboptimal performance. In contrast, our approach is designed to leverage both the hierarchical architecture and the two-level alignment loss.

## 3 HIERARCHICAL PART-TO-WHOLE VISION-LANGUAGE ALIGNMENT

We introduce F-CAST, a novel vision-language model that learns hierarchical part-to-whole alignment from long captions. It combines F̲LAIR and C̲AST, leveraging CAST's part-whole struc-

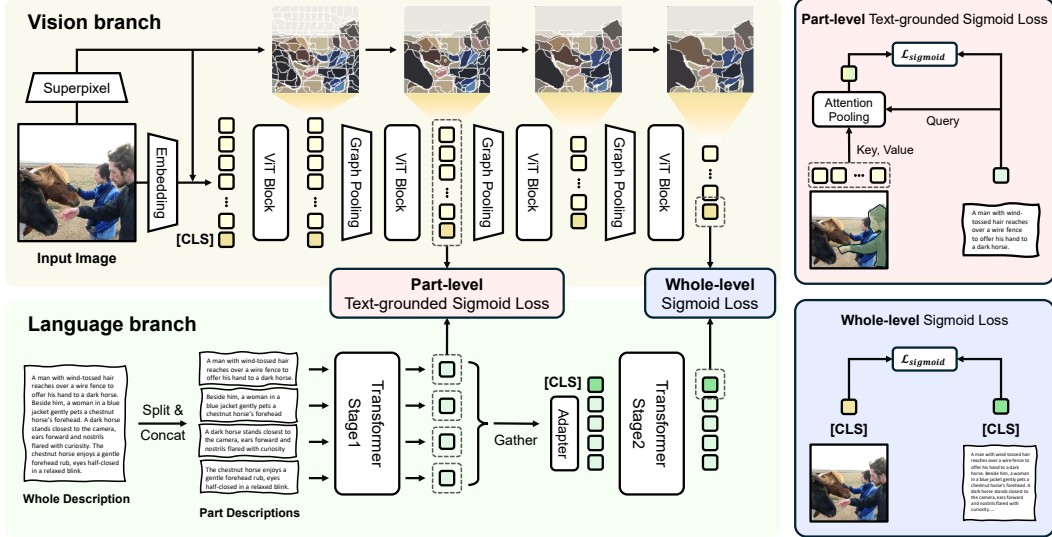

Figure 5: **Overview of our proposed F-CAST.** The model builds hierarchical representations in both vision and language and aligns them at matched granularities. **Visual hierarchy**: We employ CAST (Ke et al., 2022) as the vision encoder to produce fine-to-coarse segment tokens via graph pooling. **Textual hierarchy**: A hierarchical transformer text encoder encodes part descriptions and a whole-caption representation (via [CLS]); an adapter gathers part tokens. **Losses**: We apply a part-level text-grounded sigmoid loss between segment tokens and their associated subcaption, and a whole-level sigmoid loss between the image embedding and the whole caption.

ture (Ke et al., 2022) to extend FLAIR's learning framework (Xiao et al., 2025). We now detail how F-CAST aligns the 1) visual hierarchy with 2) long caption hierarchy 3) in a part-to-whole manner.

### 3.1 PART-WHOLE STRUCTURE IN VISION

To understand the complex visual scenes, people parse them into part-whole hierarchies (Hinton, 1979), providing *what-is-where* substantiation and specifying compositional structure. Our objective in the vision branch is learning such an explicit part-whole hierarchy for recognition and by recognition.

We build upon recent work CAST (Ke et al., 2022). CAST employs superpixel tokens that are progressively merged via the proposed graph pooling module. Their fine-to-coarse superpixel grouping, learned under the recognition objective, naturally reveals part-whole relationships as an internal part of the recognition process (see the top of Figure 11).

Specifically, our model starts from 196 superpixel tokens and progressively groups them into 64, 32, and 16, thereby moving from finer to coarser segment tokens. Among these, the segment tokens at the 2nd stage, denoted as $\mathbf{v}^{\text{fine}} \in \mathbb{R}^{64 \times C}$, are used for the part-level alignment, while the [CLS] token from the final layer, denoted as $\mathbf{v}^{\text{coarse}} \in \mathbb{R}^C$, serves as the representation for the whole-level alignment. This selection is for matching grounding granularity. The lower level of the hierarchy contains more localized and finer-grained information while the higher levels capture more holistic semantics. Therefore, finer segment features can be effective in capturing fine details for object description (e.g., *red double-decker bus, specifically labeled route 38*), whereas coarser segment features can be helpful for scene description.

### 3.2 PART-WHOLE STRUCTURE IN LANGUAGE

Aligning the visual hierarchy with the textual hierarchy in a one-to-one manner is highly challenging. Whether the textual hierarchy is semantic or syntactic, elements that are close in visual appearance may be far apart in textual meaning, and vice versa. However, from long captions we identify a new *spatial hierarchy* that aligns well with the visual hierarchy. Previous work (Zheng et al., 2024) observed that each sub-caption of a long caption tends to describe a specific part of

the scene (e.g., an object, a part of an object, or background context). We therefore interpret the hierarchy between sub-captions and the long caption as corresponding to the part–whole structure of the image. To effectively capture the hierarchical structure where sub-captions compose a long caption, we propose a *hierarchical transformer* text encoder, with a divide-and-conquer manner: (1) first stage encodes each sub-captions *independently* to learn a distinct representation, and (2) second stage encodes the long caption from pre-generated sub-caption embedding from the first stage.

**Chunking.** We first obtain sub-captions by splitting the long caption into $N$ chunks $T_1, T_2, ..., T_N$: we split the original caption into individual sentences following DreamLIP (Zheng et al., 2024), and then concatenate 1-3 consecutive sentences into each chunk. See more details in the Section C.

**Transformer Stage 1.** Next, we independently forward each chunk through a stage-1 transformer to encode $N$ sub-caption embeddings $\mathbf{t}^{\text{sub}}$.

$$\mathbf{t}_k^{\text{sub}} = \text{TransformerStage1}(\text{Tokenize}(T_k)) \in \mathbb{R}^D, \quad k = 1, \ldots, N \tag{1}$$

$$\mathbf{t}^{\text{sub}} = \{\mathbf{t}_k^{\text{sub}}\}_{k=1}^N \in \mathbb{R}^{N \times D} \tag{2}$$

**Transformer Stage 2.** Finally, we obtain the long caption embedding $\mathbf{t}^{\text{long}} \in \mathbb{R}^D$ by feeding the sub-caption embeddings together with a [CLS] token into the stage-2 transformer. Before we feed, we refine the sub-caption embeddings with light-weight residual MLP adapter (Gao et al., 2024):

$$\mathbf{t}^{\text{long}} = \text{TransformerStage2}\big(\text{Adapter}(\mathbf{t}^{\text{sub}}); \texttt{[CLS]}\big) \in \mathbb{R}^D \tag{3}$$

### 3.3 Two-level Vision-Language Alignment

**Training batch construction**. We train F-CAST from scratch on a synthetic long text-image paired dataset, where each image is annotated with multiple long and short captions. At each iteration, we construct a batch of $B$ images, each image $I_i$ paired with $K$ positive sub-captions $\{T_{i_k}\}_{k=1}^K$ Among the $K$ sub-captions, the first $N$ sub-captions are derived from a single long caption, while the remaining $(K-N)$ sub-captions are sampled from multiple captions. Therefore, the second stage of the text encoder takes only the first $N$ sub-captions for its input. The resulting batch input and hierarchical outputs are therefore constructed as follows:

$$\text{batched\_input} : \{(I_i, \{T_{i_k}\}_{k=1}^K)\}_{i=1}^B, \tag{4}$$

$$\text{intermediate\_outputs} : \{(\mathbf{v}_i^{\text{fine}}, \{\mathbf{t}_{i_k}^{\text{sub}}\}_{k=1}^K)\}_{i=1}^B, \quad \text{final\_outputs} : \{(\mathbf{v}_i^{\text{coarse}}, \mathbf{t}_i^{\text{long}})\}_{i=1}^B \tag{5}$$

**Part-level Text-grounded Sigmoid Loss** aim to align each sub-caption feature with its corresponding visual region features without dense annotation (e.g., bounding box). Following FLAIR, we adopt an attention pooling module (Ilse et al., 2018) to obtain text-grounded visual features. Let $\mathbf{v}_i^{\text{fine}}$ denote visual segment features from image $i$, and $\mathbf{t}_{j_k}^{\text{sub}}$ denote $k$-th sub-caption features from caption $j$. Then attention pooling computes a **t**ext-**g**rounded visual feature:

$$\mathbf{v}_{i,j_k}^{\text{tg}} = \text{AttnPool}\big(\mathbf{t}_{j_k}^{\text{sub}}, \mathbf{v}_i^{\text{fine}}, \mathbf{v}_i^{\text{fine}}\big) \in \mathbb{R}^D \tag{6}$$

This operation aggregates the visual segments relevant to the given sub-caption query $\mathbf{t}_{j_k}^{\text{sub}}$ via attention scores. We then compare $\mathbf{v}_{i,j_k}^{\text{tg}}$ again with the query $\mathbf{t}_{j_k}^{\text{sub}}$:

$$\mathcal{L}_{i,j,k}^{\text{part}} = \frac{1}{1 + \exp\Big( y_{i,j} \big(-\tau_{\text{part}}\langle \mathbf{v}_{i,j_k}^{\text{tg}}, \mathbf{t}_{j_k}^{\text{sub}}\rangle + b_{\text{part}}\big)\Big)}. \tag{7}$$

where $\tau_{\text{part}}$ and $b_{\text{part}}$ is a learnable temperature and a bias term, and $\langle \cdot, \cdot \rangle$ denotes cosine similarity. The label $y_{i,j}$ indicates whether the pair is postive or negative: $y_{i,j}{=}{+}1$ for a positive pair ($\mathbf{v}_{i,i_k}^{\text{tg}}$, $\mathbf{t}_{i_k}^{\text{sub}}$), and $y_{i,j}{=}{-}1$ for a negative pair (when $i \neq j$).

**Whole-level Sigmoid Loss** aims to align the global image feature $\mathbf{v}_i^{\text{coarse}}$ with the long caption feature $\mathbf{t}_i^{\text{long}}$, in contrast to FLAIR, which aligns the global image feature with multiple sub-captions.

$$\mathcal{L}_{i,j}^{\text{whole}} = \frac{1}{1 + \exp\Big( y_{i,j} \big(-\tau_{\text{whole}}\langle \mathbf{v}_i^{\text{coarse}}, \mathbf{t}_j^{\text{long}}\rangle + b_{\text{whole}}\big)\Big)}. \tag{8}$$

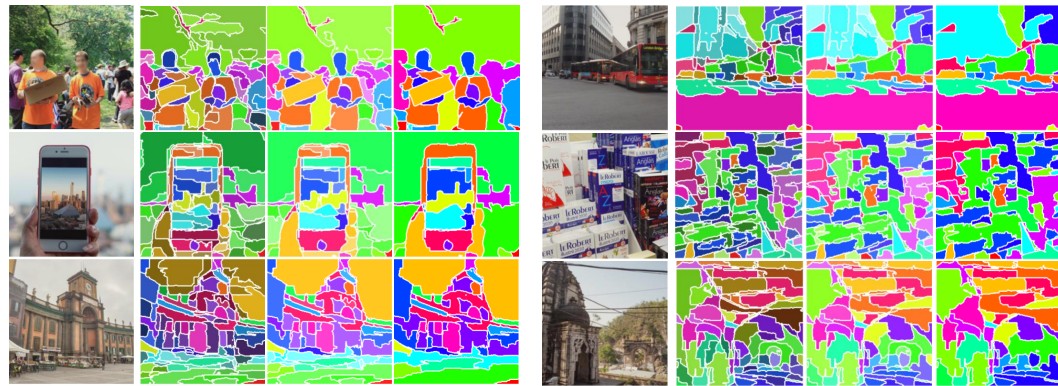

<table>
<tr><td>(a) Retreival Correct Predictions</td><td>(b) Retreival Incorrect Predictions</td></tr>
</table>

Figure 6: **Visual part-to-whole parsing quality correlates with image-text retrieval.** (a) **Correct** predictions exhibit coherent fine-to-coarse grouping that uncovers explicit part-to-whole struction. (b) **Incorrect** predictions show fragmented, inconsistent groupings that correlate with misalignment. We compare fine-to-coarse segmentations for correct (a) and incorrect (b) image-to-text retrieval results on the Urban-1k and ShareGPT4V datasets to illustrate how the quality of hierarchical grouping affects comprehensive scene understanding.

We use separate learnable parameters for the whole-level sigmoid loss: $\tau_{\text{part}}$ and $b_{\text{part}}$, distinct from the part-level ones. We use the same $y_{i,j}$ as in part-level one.

**Optimization.** We optimize F-CAST with two proposed loss functions: $\mathcal{L} = \mathcal{L}^{\text{part}} + \mathcal{L}^{\text{whole}}$. The $\mathcal{L}^{\text{part}}$ encourages alignment between the visual region and sub-captions at the intermediate layer, while the $\mathcal{L}^{\text{whole}}$ enforces alignment between the entire image and long caption at the final layer. We adopt a sigmoid loss instead of softmax for both $\mathcal{L}^{\text{part}}$ and $\mathcal{L}^{\text{whole}}$. Sigmoid is not only more stable when the batch size is small but also effective at dealing with multiple positives (Zhai et al., 2023).

**Inference with F-CAST.** F-CAST only utilizes whole-level alignment at inference time. As the bottom-up hierarchy sufficiently enriches the holistic embeddings, making part-level alignment unnecessary. In other words, part-level alignment can be viewed as an auxiliary loss that boosts fine-grained and localized image embedding for whole understanding.

## 4 EXPERIMENTS

We study whether F-CAST discovers better visual parts and wholes (Section 4.2) and sub-caption grounding (Section 4.3), and how it performs on image-text retrieval benchmarks (Section 4.4). We then analyze contributions of our design choices (Section 4.5).

### 4.1 EXPERIMENTAL SETTING

**Training Datasets.** We train F-CAST on DreamLIP's (Zheng et al., 2024) re-captioned datasets, which provide long and detailed captions sufficiently for learning both part and whole-level visual representations. The DreamLIP dataset consists of CC3M-recap, CC12M-recap, and YFCC15M-recap, which are merged into a unified corpus of 30M samples (Merged-30M).

**Implementation Details.** We develop our model based on FLAIR (Xiao et al., 2025) code implementation. We employ CAST-B (Ke et al., 2022) as the vision encoder, which is comparable in size to ViT-B/16 (Dosovitskiy et al., 2020). The input image size is set to $224 \times 224$. We follow the vanilla Transformer architecture (Vaswani et al., 2017) for the text encoder, except for adopting a two-stage design. The first stage transformer consists of $L_1 = 8$ layers, while the second stage has $L_2 = 4$ layers. In total, this results in $L = 12$ layers, matching the depth of the original CLIP text encoder (Radford et al., 2021). The context length of the first stage is set to 77, consistent with CLIP, whereas the second stage uses a context length of $N = 4$, corresponding to the chunk size. The embedding dimension is set to 512 for both image and text features across our study. For a fair comparison with FLAIR, we follow their training hyperparameter but use a reduced batch size of

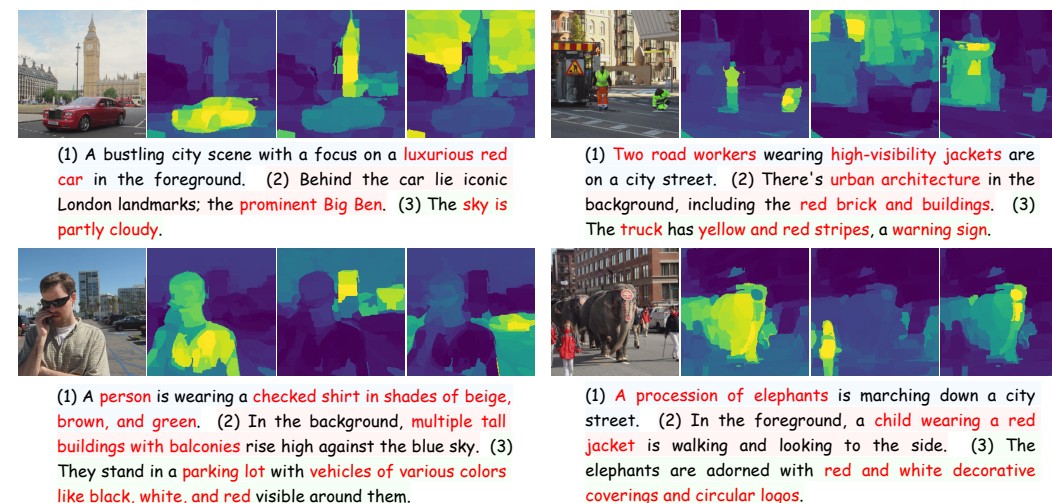

Figure 7: **F-CAST provides spatially precise, compositional grounding.** We show head-averaged attention maps from the AttnPool module for three subcaptions per example. The leftmost image is the input; the three heatmaps to the right depict attention for subcaptions (1) to (3). Our attention maps localize precisely to relevant regions across spatial positions (*worker* vs. *building*) and scales (*nearby* vs. *distant*), with sharp boundaries that reflect a consistent visual hierarchy. Interestingly, the attention also exhibits compositional behavior. For instance, in the *behind the car lies the prominent Big Ben* example (top left), both the *car* and the *Big Ben* are activated, with stronger focus on Big Ben, reflecting its greater prominence in the caption. Other examples include *road worker with high-visibility jacket* (top right) and *elephants adorned with coverings* (bottom right).

2K due to GPU memory limits. We also follow the number of subcaptions per image, $K=8$. The learnable temperature parameters $\tau_{part}$, $\tau_{whole}$ and bias terms $b_{part}$, $b_{whole}$ are initialized to 0.07 and -10, respectively, following SigLIP (Zhai et al., 2023). We utilized 8 A100 GPUs (80GB) for training. The training duration scaled linearly with the dataset size, and training Merged-30M required 5 days (total 40 GPU-days).

## 4.2 PART-WHOLE HIERARCHY IN VISION

Our vision encoder learns a part-to-whole hierarchy via progressive fine-to-coarse token merging. This inductive bias composes small, locally coherent regions into progressively larger structures and reveals how the model recognizes a scene (Figure 6). Notably, the hierarchy emerges under language-only supervision (i.e., without any segmentation labels). As shown in Figure 6a, the encoder naturally aggregates micro-parts into semantically meaningful units (e.g., fingers into a hand).

This hierarchy is not merely interpretable; it is essential for comprehensive scene understanding. When grouping is correct, evidence from parts integrates into a coherent whole, yielding faithful image-text alignment and accurate retrieval. When grouping is incorrect, local cues do not compose, and the model fails to retrieve the ground-truth image (e.g., fails to parse individual books in Figure 6b). Therefore, explicit fine-to-coarse grouping is key to achieving both fine-grained detail and global coherence, enabling robust scene comprehension under language-only supervision.

## 4.3 VISUAL GROUNDING PROPERTY

Figure 7 visualizes head-averaged attention maps from the AttnPool module for each subcaption, illustrating how F-CAST aggregates relevant segment tokens. In the first example, the model grounds not only the target object (i.e., car) but also its background context (e.g., Big Ben and sky), indicating compositional understanding. The road-workers case demonstrates that F-CAST is able to localize both large structures (e.g., buildings) and small objects (e.g., workers). The elephant example shows that the model grounds both the entire object (e.g., elephant) and its specific parts (e.g., logos).

These behaviors emerge at an intermediate encoder layer, where localized semantics are formed and subsequently merged into a holistic scene representation. Taken together, the visualizations

Table 1: **F-CAST achieves state-of-the-art results on zero-shot long text-image retrieval tasks.** I2T and T2I indicate the R@1 score on image-to-text and text-to-image retrieval, respectively. The best results are **bold**, second-best are underlined. All models use ViT-B/16 as the vision encoder. DCI and DOCCI use human-annotated captions, while the other datasets rely on VLM-generated synthetic captions and generally achieve higher scores. F-CAST consistently outperforms prior methods across all benchmarks by large margins.

| Method | Data | DCI | | DOCCI | | SV-1k | | SV-10k | | Urban-1k | | IIW | |
|---|---|---|---|---|---|---|---|---|---|---|---|---|---|
| | | I2T | T2I | I2T | T2I | I2T | T2I | I2T | T2I | I2T | T2I | I2T | T2I |
| *Trained on Short-Captions Only* | | | | | | | | | | | | | |
| OpenCLIP | 2B | 56.0 | 55.4 | - | - | 90.3 | 87.7 | 69.6 | 66.8 | 69.5 | 65.8 | - | - |
| LiT | 100M | 41.7 | 40.9 | - | - | 86.0 | 80.0 | 61.4 | 50.6 | - | - | - | - |
| ALIGN | 700M | 56.5 | 57.4 | - | - | 86.3 | 85.3 | 65.1 | 62.7 | - | - | - | - |
| SigLIP | 10B | 57.7 | 56.2 | - | - | 85.8 | 83.4 | 83.4 | 63.0 | 62.7 | 62.1 | - | - |
| *Trained on Short Captions → Finetuned on Long-Captions* | | | | | | | | | | | | | |
| Long-CLIP | 400M→1M | 47.4 | 44.1 | - | - | 90.6 | 87.4 | 73.1 | 62.0 | 78.9 | 79.5 | - | - |
| FineLIP | 400M→1M | - | - | 77.1 | 79.5 | - | - | - | - | 90.7 | 89.3 | - | - |
| TULIP | 400M→1M | - | - | - | - | 98.6 | 98.6 | - | - | 88.1 | 86.6 | - | - |
| FG-CLIP | 400M→1.6B | 61.8 | 60.6 | - | - | 96.7 | 94.9 | - | - | - | - | - | - |
| *Trained on Long-Captions from Scratch* | | | | | | | | | | | | | |
| LoTLIP | 100M | 62.1 | 61.0 | - | - | 95.5 | 86.8 | 86.8 | 81.4 | - | - | 94.0 | 92.5 |
| FLAIR | 30M | 61.3 | 66.2 | 70.3 | 72.6 | 98.5 | 98.0 | 90.3 | 89.4 | 83.6 | 87.7 | 91.3 | 91.5 |
| **F-CAST (ours)** | 30M | **69.4** | **69.4** | **80.6** | **80.4** | **99.0** | **98.8** | **95.1** | **94.5** | **93.6** | **93.9** | **97.4** | **96.1** |
| *vs. previous SOTA* | | +7.3 | +3.2 | +3.5 | +0.9 | +0.4 | +0.2 | +4.8 | +5.1 | +2.9 | +4.6 | +3.4 | +3.6 |

indicate that F-CAST delivers spatially precise, compositional grounding that supports robust scene understanding. Additional visualizations are available in Appendix F.

## 4.4 ZERO-SHOT LONG TEXT-IMAGE RETRIEVAL ON BENCHMARK

**Evaluation Benchmark.** We evaluate Recall@1 (R@1) for long text-image cross-modal retrieval on six widely used benchmarks: DCI (Urbanek et al., 2024), DOCCI (Onoe et al., 2024), ShareGPT4V-1k (Chen et al., 2024), ShareGPT4V-10k (Chen et al., 2024), Urban-1k (Zhang et al., 2024), and IIW (Garg et al., 2024). Detailed statistics for each dataset are provided in Appendix D.

**Comparison with state-of-the-art.** We organize prior work into three settings: i) models trained only on short captions, ii) models pretrained on short captions and then finetuned on long captions, and iii) models trained from scratch on long captions. Table 1 provides an extensive comparison across six long-text benchmarks (DCI, DOCCI, SV-1k, SV-10k, Urban-1k, and IIW), while most existing studies evaluate on only two or three datasets. Within this unified setting, F-CAST achieves the best overall performance, surpassing FLAIR at the same data scale (30M pairs) and outperforming finetuned models despite using far fewer pretraining samples. This highlights both the effectiveness and the data efficiency of our hierarchical design for long-caption retrieval.

**Scaling behavior.** As shown in Figure 8, both FLAIR and F-CAST improve steadily as the training data scales from 3M to 30M pairs. F-CAST consistently stays above FLAIR with a similar slope, showing that part–whole alignment is compatible with scaling laws and improves data efficiency at small scales while remaining effective as data grows.

## 4.5 ABLATION STUDIES: FROM FLAIR TO F-CAST

We perform an ablation study on CC3M-recap (batch size 2K), starting from FLAIR and examining the three components of F-CAST: i) CAST for the visual hierarchy, ii) hierarchical transformer for the textual hierarchy, and iii) a two-level alignment loss for hierarchical alignment. Table 2 shows that all three components contribute, demonstrating the necessity of aligning visual and textual hierarchies.

**Visual hierarchy.** Leveraging CAST, we observe consistent performance gains across all datasets (second row in Table 2). This improvement stems from the hierarchical structure introduced by CAST. A flat ViT partitions images into uniform patches without explicit structure, whereas CAST

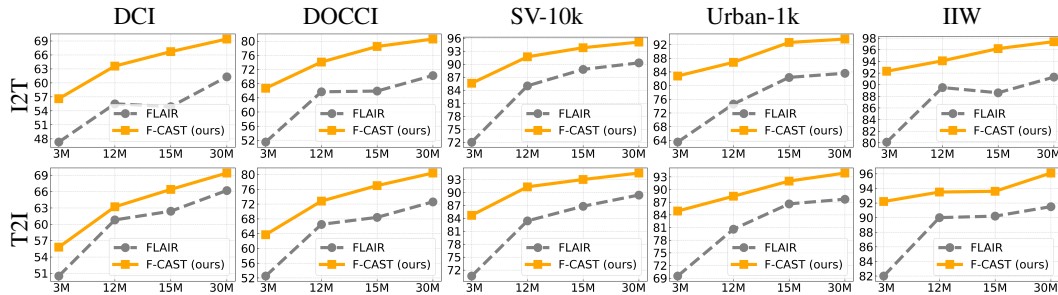

Figure 8: **Scaling laws for FLAIR and F-CAST.** F-CAST consistently achieves higher accuracy than FLAIR as the training set grows, with both models improving steadily as the scale increases. This suggests that our hierarchical inductive bias enables robust scaling behavior.

Table 2: **Ablation study from FLAIR to F-CAST.** Relative to FLAIR, F-CAST introduces three changes: i) replacing the ViT vision encoder with CAST, ii) replacing the CLIP text encoder with hierarchical transformer, and iii) adding the two-level alignment loss. All three components contribute to the overall performance gains, together leading to the final F-CAST model.

| Method | DCI | | DOCCI | | SV-10k | | Urban-1k | | IIW | |
|---|---|---|---|---|---|---|---|---|---|---|
| | I2T | T2I | I2T | T2I | I2T | T2I | I2T | T2I | I2T | T2I |
| FLAIR (reproduced) | 48.3 | 52.0 | 52.9 | 53.6 | 73.3 | 72.1 | 67.3 | 73.7 | 80.4 | 83.5 |
| + ViT → CAST | 49.8 | 54.1 | 54.0 | 55.0 | 76.1 | 74.7 | 71.3 | 73.8 | 82.5 | 83.8 |
| + CLIP text encoder → hierarchical | 53.4 | 50.0 | 63.5 | 58.9 | 82.3 | 82.4 | 78.6 | 81.8 | 91.0 | 87.3 |
| + flat loss → two-level loss | **56.6** | **55.8** | **66.7** | **63.7** | **85.6** | **84.7** | **82.8** | **84.9** | **92.3** | **92.2** |

merges superpixels into coarser segments, forming a natural hierarchy from parts to wholes. This inductive bias is crucial because sub-captions typically describe objects or regions that require part-level grounding, which is difficult without a compositional visual encoder.

**Textual hierarchy.** In our ablations, hierarchical transformer for text encoder yields the largest single performance gain (third row), suggesting that language modeling may have a stronger impact than visual in long-text image pre-training. This improvement arises from the way the hierarchical transformer handles long caption. whereas CLIP's text encoder flattens long captions, conflating local and global semantics, our hierarchical transformer first encodes each sub-caption and then composes them into a whole-caption representation, preserving local detail while supporting holistic understanding. This hierarchical encoding complements the visual hierarchy and enables long captions to serve as effective supervision for fine-grained grounding. Furthermore, our hierarchical text encoder enables processing of longer inputs without being truncated by the 77-token context limit.

**Two-level alignment loss.** Compared with a flat loss, the two-level loss consistently leads to significant performance gains (fourth row). Even with hierarchical encoders, parts and wholes should not be learned in isolation. The two-level cross-modal alignment loss explicitly links local alignments to global ones, ensuring that detailed matches integrate into a coherent scene representation.

## 5    CONCLUSION

F-CAST is a hierarchical vision-language framework that aligns the visual hierarchy of images with the spatial hierarchy of long captions. It combines a CAST-based vision encoder for fine-to-coarse visual grouping with a hierarchical transformer text encoder that encodes subcaptions and fuses them into a whole-caption representation. A two-level cross-domain alignment loss links these hierarchies: mid-level segment tokens align with sub-caption embeddings for localized grounding, while top-level tokens align with the whole-caption representation. Trained on 30M long–text-image pairs, F-CAST achieves state-of-the-art performance on six long-text retrieval benchmarks, exhibits strong scaling and data efficiency.

**Ethics statement.** This work adheres to the ICLR Code of Ethics. The research does not involve human or animal subjects, nor does it rely on private or sensitive data. All datasets used are publicly available, and we comply with their licenses and usage guidelines. To the best of our knowledge, the methods and findings do not pose foreseeable risks of harm, misuse, or negative societal impact.

**Reproducibility statement.** Implementation details are provided in sections 3 and 4, and the Appendix. The full code will be released upon acceptance.

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

# Aligning Forest and Trees in Images and Long Captions for Cross-Domain Grounding

## Supplementary Material

C​ONTENTS

## A   TRAINING CONFIGURATIONS

We follow FLAIR's pretraining configuration as displayed in Table 3. However, we use 2K batch size for all datasets, due to GPU RAM limit.

Table 3: Training configuration for different datasets.

| Config | CC3M-recap | CC12M-recap | YFCC15M-recap | Merged-30M |
|---|---|---|---|---|
| Batch size | | 2,048 | | |
| Optimizer | | AdamW (Loshchilov & Hutter, 2017) | | |
| Learning rate | | $5 \times 10^{-4}$ | | |
| Weight decay | 0.5 | 0.5 | 0.5 | 0.2 |
| Adam $\beta$ | | $\beta_1, \beta_2 = (0.9, 0.98)$ | | |
| Adam $\epsilon$ | $1 \times 10^{-8}$ | $1 \times 10^{-8}$ | $1 \times 10^{-8}$ | $1 \times 10^{-6}$ |
| Total epochs | | 32 | | |
| Warm up | | 2,000 (steps) | | |
| LR scheduler | | cosine decay | | |

## B   DIRECT COMPARISON WITH CAST

Compared to naïve CAST, we differ in three aspects. First, we further explore language supervision as the recognition objective, whereas CAST only considers discrimination (Chen et al., 2021) or label supervision. Second, we encourage segment tokens to directly localize semantics via part-level loss, unlike CAST, which handles supervision only at the global level (i.e., [CLS] token). Third, we replace CAST's superpixel algorithm SEEDS (Van den Bergh et al., 2012) with SFCN (Yang et al., 2020), which alleviates its limitation in capturing thin structures (see the tree branch in the first image of (a) in Figure 6). Additionally, we discard the final graph pooling module in CAST (reducing 16 tokens to 8), since it is not used during the pre-training stage.

## C   DETAILS ON THE CHUNKING STRATEGY

In Section 3.2, we briefly introduced the idea of dividing a long caption into $N$ sub-captions (chunks). This section provides a detailed description of the chunking strategies used in training and inference. Specifically, we adopt **random chunking** during training and **balanced chunking** during inference.

Chunking begins by splitting a long caption into $L$ sentences:

$$S_1, S_2, \ldots, S_L,$$

where $L$ is a variable length depending on each sample.

**Random chunking.** During training, each chunk is formed to contain 1, 2, or 3 sentences, with the number chosen randomly. We enforce two constraints: (1) each chunk must contain at least one sentence, and (2) no chunk may contain more than three sentences. Therefore, when $L > 3N$, excess sentences are discarded. Conversely, if $L < N$, sentences are randomly resampled with replacement until all chunks are filled.

**Balanced chunking.** During inference, results may vary depending on how sentences are distributed across chunks. To ensure stable and reproducible inference, we employ balanced chunking, which divides sentences as fairly as possible. Each chunk first receives $\lfloor L/N \rfloor$ sentences, and the remaining sentences are allocated to the earlier chunks in order.

For example:

$$\text{if } L = 6, N = 4 \;\Rightarrow\; [2, 2, 1, 1],$$
$$\text{if } L = 11, N = 4 \;\Rightarrow\; [3, 3, 3, 2].$$

It is worth noting that our divide-and-conquer approach with chunking is also computationally efficient under the quadratic self-attention mechanism (Song et al., 2024). In contrast, recent CLIP

variants for long-text understanding (Zhang et al., 2024; Wu et al., 2024; Asokan et al., 2025; Najdenkoska et al., 2024) process up to 248 tokens at once, leading to substantial computational cost.

# D    STATISTICS ON THE LONG-TEXT IMAGE RETRIEVAL BENCHMARKS

We present the detailed statistics of the long-text-image retrieval datasets we use in Table 4. For DOCCI, we use only the test split for evaluation, leaving the rest untouched.

Table 4: Statistics of the long-text-image retrieval datasets.

| Dataset | # Images | # Texts | # Sub-captions per Text | # Tokens per Text |
|---|---|---|---|---|
| DCI | 7,805 | 7,805 | 10.81 | 172.73 |
| DOCCI | 5,000 | 5,000 | 7.12 | 141.52 |
| ShareGPT4v-1k | 1,000 | 1,000 | 8.15 | 173.24 |
| ShareGPT4v-10k | 10,000 | 10,000 | 8.24 | 173.66 |
| Urban-1k | 1,000 | 1,000 | 5.97 | 131.36 |
| IIW | 612 | 612 | 10.16 | 239.73 |

# E    ADDITIONAL ABLATIONS

In this section, we present an additional ablation study on the role of two-level losses. In Section 4.5, we compared our hierarchical loss with FLAIR's flat loss, where both whole-level and part-level alignments are applied only at the top layer. Another design choice, adopted by LoTLIP (Wu et al., 2024), is to use only the whole-level alignment loss at the top layer. Table 5 reports the comparison between our full F-CAST and a variant without the part-level alignment loss. Removing the part-level loss leads to a significant performance drop, directly demonstrating that fine-grained scene understanding requires detailed associations of local elements.

# F    ADDITIONAL VISUALIZATIONS

We present additional attention maps from the $\mathrm{AttnPool}$ module for each subcaption in Figure 9. The visualizations illustrate how F-CAST grounds colors, shapes, and text in images. As a sanity check, the attention map remains inactive when a caption does not correspond to the image.

# G    THE USE OF LARGE LANGUAGE MODELS (LLMS)

Large Language Models (LLMs) were used solely as general-purpose editing tools to refine grammar and phrasing. They did not contribute to research ideation, analysis, or substantive writing.

Table 5: **Ablation study on part-level alignment loss.**

| Method | DCI | | DOCCI | | SV-10k | | Urban-1k | | IIW | |
|---|---|---|---|---|---|---|---|---|---|---|
| | I2T | T2I | I2T | T2I | I2T | T2I | I2T | T2I | I2T | T2I |
| FLAIR (reproduced) | 48.3 | 52.0 | 52.9 | 53.6 | 73.3 | 72.1 | 67.3 | 73.7 | 80.4 | 83.5 |
| F-CAST (ours) | **56.6** | **55.8** | **66.7** | **63.7** | **85.6** | **84.7** | **82.8** | **84.9** | **92.3** | **92.2** |
| F-CAST w/o Part alignment | 51.4 | 54.1 | 65.2 | 60.7 | 82.1 | 82.2 | 80.7 | 83.2 | 88.4 | 86.9 |

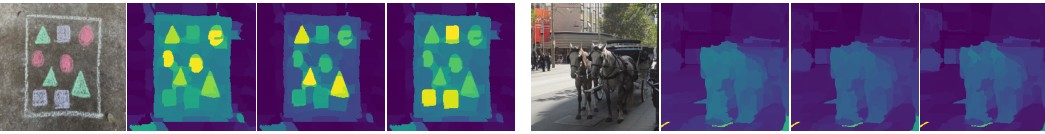

(1) Three pink circles.
(2) Three green triangles.
(3) Three light purple squares.

(1) A cute brown cat is lying on sofa.
(2) The soccer player in a white uniform is celebrating.
(3) The boy is looking at the stars through a telescope.

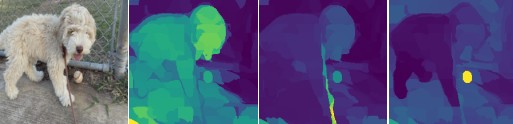

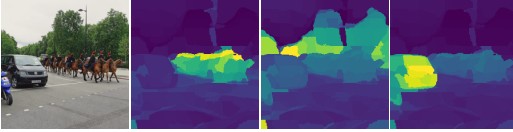

(1) A cream colored labrador puppy standing on a concrete floor.
(2) The puppy has a brown leash on.
(3) A white baseball sits in front of the fence.

(1) The image captures a procession of horse-mounted individuals, likely ceremonial guards.
(2) The scene is set against a backdrop of abundant green foliage from trees lining the street.
(3) Traffic is seen including a black Volkswagen Transporter van.

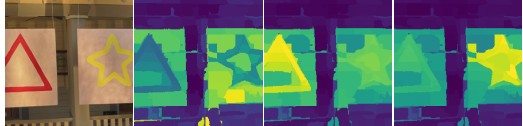

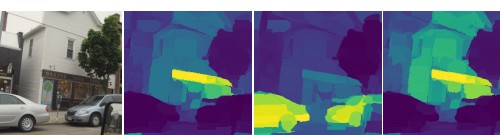

(1) Both poster boards have a patchy painted beige background.
(2) The board on the left has a triangle that is hand painted in a thick red line.
(3) The board on the right has a five-point star that is hand painted in a thick yellow line.

(1) This image captures a street view featuring the façade of a two-story building branded "ORANGE TREE IMP".
(2) In the foreground, part of a silver sedan and the front end of a gray minivan are visible.
(3) The structure has a gable roof and is painted white with a contrasting black trim around the storefront.

Figure 9: **F-CAST provides spatially grounded, compositional visual grounding.** We visualize attention maps from the AttnPool module for three subcaptions in each example. The leftmost image is the input; the three heatmaps to the right show head-averaged attention with respect to subcaptions (1)-(3). The scribble example shows F-CAST distinguishing *pink circles* from *green triangles*. With incorrect captions, F-CAST's attention maps remain mostly inactive. Querying with the text *ORANGE TREE IMP*, F-CAST grounds it accurately. These are preliminary examples, with possible confounding factors.

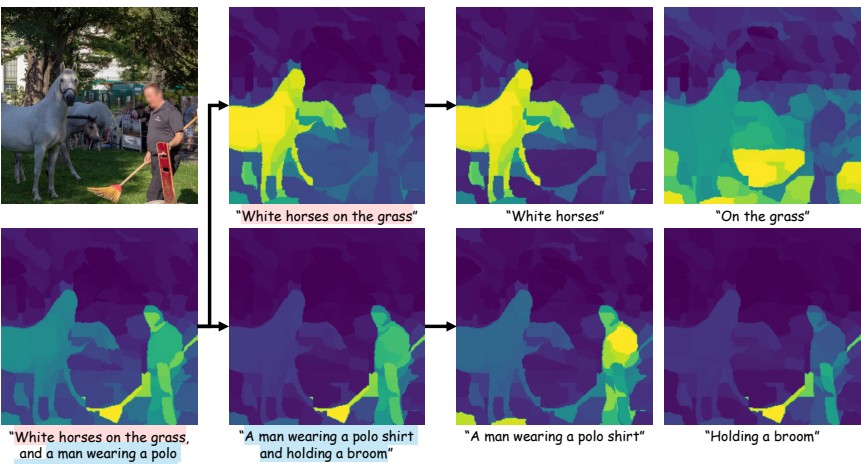

Figure 10: **F-CAST is capable of visual grounding across multiple levels of granularity**. A single caption may describe different objects simultaneously, and can be decomposed into smaller units. To reflect this hierarchical nature of caption, we decompose the sentence-level caption into smaller phrases and visualize their corresponding grounding results. Remarkably, F-CAST (1) can locate multiple objects at the same time (e.g., both 'a man' and 'horses' in the bottom left) and (2) precisely identify individual parts in the phrases (e.g., 'holding a broom' in the bottom right).

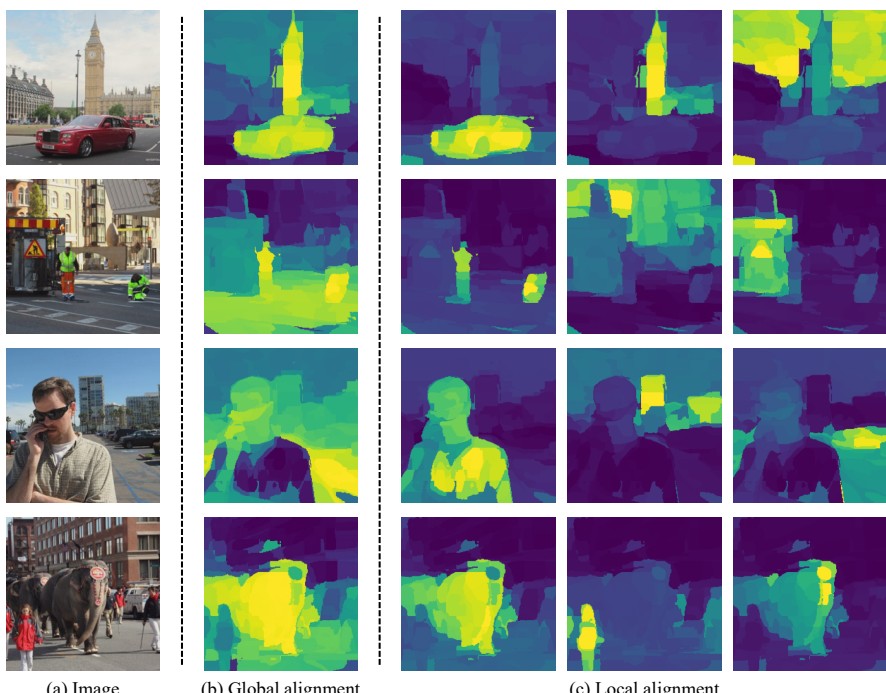

(a) Image  (b) Global alignment  (c) Local alignment

Figure 11: **Visualization of global alignment**. We revisit the examples from Fig. 7 and compare global alignment, captured by the full long caption, with local alignment derived from each sub-caption. As shown in (b), the global alignment results based on CAM (Li et al., 2025) highlight broad, major regions of the images, whereas each local alignment in (c) focuses on more specific and localized areas.

