# OpenReview forum: "Aligning Forest and Trees in Images and Long Captions for Cross-Domain Grounding"
_ICLR.cc/2026/Conference — Submitted to ICLR 2026_

### Official Review · Reviewer_frwe · 2025-10-29

**Soundness:** 3
**Presentation:** 2
**Contribution:** 2
**Rating:** 4
**Confidence:** 3

**Summary:**

The paper proposes F-CAST, a hierarchical vision–language pretraining framework for long captions. On the vision side, it replaces a flat ViT with CAST to produce fine-to-coarse **segment tokens.** On the text side, it uses a two-stage hierarchical transformer: **Stage-1** encodes sub-captions (sentences/chunks), **Stage-2** composes them (with an adapter) into a whole-caption embedding. Cross-modal training uses two sigmoid losses: (i) a part-level text-grounded loss that aligns attention-pooled visual segments with sub-captions, and (ii) a whole-level loss aligning a global image token with the composed whole-caption embedding.

**Strengths:**

* Unified hierarchical alignment with simple losses; easy to slot into CLIP-style training while improving long-text robustness.

* Strong empirical gains across six benchmarks; scaling plot indicates gains persist with more data.

* Ablations indicate each piece (CAST, hierarchical text, two-level loss) contributes materially.

**Weaknesses:**

* **Limited novelty— Most ingredients are adaptations of prior work:** CAST for the visual hierarchy, and FLAIR-style text-conditioned image representations extended to a two-level (part/whole) loss. The paper itself positions F-CAST as “adopt CAST” + “extend FLAIR” with hierarchical text and a two-level alignment objective, which reads more like a careful systemization than a fundamentally new algorithmic idea.



* **Grounding validation gap:** relies on attention maps; lacks quantitative phrase↔region grounding benchmarks, making the “discovered hierarchy” claim hard to verify scientifically.


* **Auxiliary-only parts at test time:** The part-level pathway is unused during inference, so is it necessary for the reported retrieval gains, or is it merely a training prior? A controlled experiment removes part-level loss at train time, but a matching compute is needed.



* **Synthetic caption dependence:** trained on long synthetic recaps (DreamLIP); no robustness audits vs. noise/hallucination or transfer to human-written dense captions beyond the ones reported.

**Questions:**

* Quantify grounding: Can you report standard grounding metrics (e.g., phrase localization/pointing accuracy or region retrieval) on datasets with region annotations (RefCOCO/RefCOCO+/Flickr30k Entities) to substantiate the part-level alignment claim? How does F-CAST compare to FLAIR under identical protocols?



* Ablate test-time dependence: What happens if you retain the part-level branch at inference (e.g., late fusion or re-ranking) vs. your current whole-only inference? Conversely, if you remove L_part during training, holding compute constant, how much of Table 1 survives? (Your Table 5 variant still shows a gap, but more controls are needed.)



* Data realism & cost: Provide robustness studies to caption noise (shuffle/perturb chunks), and report FLOPs/throughput/memory vs. FLAIR/Long-CLIP to justify practicality. Also, clarify exact GPU-days per dataset and training schedule details beyond “5 days on 8×A100 for 30M”.


* Where does part-level alignment help? Your inference uses only the global embedding. What specific cases benefit from the part-level loss?


**I am open to changing my score based on the author's responses.**

---

> ### Author Response · Authors · 2025-11-21
>
> Dear reviewer frwe,
>
> Thank you for your thoughtful and detailed feedback. We appreciate your recognition of our unified hierarchical alignment, the strength of our empirical results, and the substantive contributions of each component. In what follows, we provide clarifications and new experiments that address each of your questions and concerns.
>
> ---
>
> **[W1] Novelty and contributions**
>
> >  "Most ingredients are adaptations of prior work … which reads more like a careful systemization than a fundamentally new algorithmic idea."
>
> Our core contribution is to introduce and realize a new paradigm for vision-language models (VLMs): the **joint alignment of visual hierarchy and textual hierarchy**. Prior works have either relied on flat textual encoders or used syntactic parse trees only on the text side. To the best of our knowledge, no prior work has explored representing **both** images and texts as hierarchical structures and **aligning the two hierarchies in a spatially oriented manner**.
>
> We show that this cross-modal hierarchical alignment becomes especially important when training with longer and more expressive captions. Our experiments demonstrate that this alignment substantially improves long-caption understanding (Table 1) and enables precise visual grounding where visual segments correspond to the textual subtrees (Figure 7).
>
> Importantly, our novelty does not lie in combining existing components to enhance performance on known tasks. Instead, the contribution is conceptual. We introduce a **new architectural paradigm** based on joint hierarchical alignment, while FLAIR, CAST, or hierarchical transformers serve simply as one practical instantiation of this idea. We believe future work can adopt more advanced architectures while still following this paradigm.
>
> To avoid misunderstanding arising from how the method is presented, we will clarify the naming choice of F-CAST. We name our method F-CAST to help readers understand that our implementation leverages FLAIR and CAST as components. However, we recognize that this naming may unintentionally shift the focus toward the components rather than the conceptual contribution. We will revise the manuscript to clarify this point, avoid expressions such as “adopt CAST” or “extend FLAIR,” and explicitly emphasize that **F-CAST is only one instantiation of our novel paradigm** of aligning the forest and the trees.
>
> In summary, the novelty of our work is in 1) identifying the missing principle of cross-modal hierarchical alignment, 2) demonstrating its effectiveness for contemporary VLMs, and 3) providing a concrete implementation that validates this new perspective.

---

> > ### Author Response · Authors · 2025-11-21
> >
> > **[W4, Q3] Robustness studies on synthetic caption dependence**
> >
> > > "trained on long synthetic recaps (DreamLIP); no robustness audits … or transfer to human-written dense captions …"
> >
> > > "Provide robustness studies to caption noise …"
> >
> > We understand the reviewer’s concern regarding the use of synthetic long captions. However, we kindly note that DCI and DOCCI are human-written dense captions datasets, and our model demonstrates state-of-the-art performance on these datasets (Table 1). We also point out that most long-caption VLMs are trained on synthetic captions, since constructing a large-scale dataset of human-written long captions is extremely costly.
> >
> > Nonetheless, to directly address the reviewers’ concerns, we conducted additional robustness studies by constructing a new perturbed-caption dataset. Following [3], we apply realistic text corruptions (e.g., QWERTY-keyboard typos) to DOCCI’s captions and measure retrieval performances on them. F-CAST shows the strongest robustness to perturbed captions, outperforming LongCLIP, which is pretrained on web-scale human-written short caption data, and FLAIR, which is trained on the same dataset as ours.
> >
> > |Model|Clean I2T|Perturb I2T|Clean T2I|Perturb T2I|
> > |:-|-:|-:|-:|-:|
> > |LongCLIP|63.2|56.2|71.0|65.7|
> > |FLAIR|70.3|62.7|72.6|65.9|
> > |F-CAST|80.6|**74.3**|80.4|**73.5**|
> >
> > ---
> >
> > **[Q3] Practicality studies on FLOPS/throughput/memory and training schedule**
> >
> > >  "… report FLOPs/throughput/memory vs. FLAIR/Long-CLIP to justify practicality. Also, clarify exact GPU-days per dataset …"
> >
> > To justify practicality, we compare the computational cost of F-CAST, LongCLIP, and FLAIR on the DOCCI image-text retrieval task. We first measure the GFLOPs of the three models and demonstrate that F-CAST is the most computationally efficient. The component-wise analysis is as follows.
> >
> > **Vision Encoder**: F-CAST adopts CAST-B as its vision encoder, which progressively reduces the number of tokens from 196 $\rightarrow$ 64 $\rightarrow$ 32 $\rightarrow$ 16. This design leads to reduced computational overhead relative to ViT-B/16.
> >
> > **Text Encoder**: F-CAST employs a hierarchical text transformer that divides a long caption into four chunks, encodes them separately, and fuses the resulting embeddings in a subsequent stage. Although this introduces additional computation relative to FLAIR, it still incurs lower cost than LongCLIP, which processes captions with a token length of 248.
> >
> > **Attention Pooling**: F-CAST and LongCLIP do not use this module during performing retrieval tasks. In contrast, FLAIR must perform attention pooling over all text embeddings in the target dataset (e.g., 5K in DOCCI) for each query, leading to significant computational overhead.
> >
> > |Model|Vision Encoder|Text Encoder|Attention Pooling|Total|
> > |:-|-:|-:|-:|-:|
> > |LongCLIP|18.32|10.91|N/A|29.23|
> > |FLAIR|18.46|3.08|54.14|75.67|
> > |F-CAST|11.34|8.27|0.00|19.61|
> >
> > We subsequently measure throughput and peak GPU memory with a batch size of 1 and AMP precision on a single NVIDIA A40 GPU. We observe comparable inference throughput across all three models, indicating that F-CAST does not face practicality concerns relative to prior work while still delivering state-of-the-art retrieval performance. This similarity in throughput, despite differences in computational cost, can be attributed to two factors: (1) FLAIR’s attention-pooling operation can be effectively parallelized by trading GPU memory for speed, and (2) although F-CAST requires substantially fewer operations, the *Farthest Point Sampling* used in its graph-pooling module is not yet fully optimized at the GPU-kernel level.
> >
> > |Model|Throughput|GPU memory (peak)|
> > |:-|-:|-:|
> > |LongCLIP|14.1 imgs/s|0.35GB|
> > |FLAIR|14.5 imgs/s|1.51GB|
> > |F-CAST|14.4 imgs/s|0.9GB|
> >
> >
> > ---
> >
> > **[Q3] Exact GPU hours per dataset**
> >
> > > "Also, clarify exact GPU-days per dataset and training schedule details beyond “5 days on 8×A100 for 30M”."
> >
> > We report the exact training time and the GPU days per dataset as follows. We will include this table in the appendix to support reproducibility.
> >
> > |Data Scale|Training Hours|Total GPU hours (x8) |Total GPU Days|
> > |-:|-:|-:|-:|
> > |3M|12.33|98.64|4.11|
> > |12M|42.5|340.0|14.17|
> > |15M|59.5|476.0|19.83|
> > |30M|114.4|915.2|38.13|
> >
> > ---
> >
> > We will add these new results to the revised manuscript if they are not already included.
> >
> > We would be happy to clarify any remaining questions or concerns.
> >
> > Sincerely, Authors
> >
> > &nbsp;
> >
> >
> > [3] Qiu et al. Benchmarking Robustness of Multimodal Image-Text Models under Distribution Shift. DMLR 2024.

---

> ### Author Response · Authors · 2025-11-21
>
> **[W2, Q1] Quantitative evaluation for visual grounding**
>
> >  "… lacks quantitative phrase $\leftrightarrow$ region grounding benchmarks."
>
> > "Can you report standard grounding metrics (e.g., phrase localization … ) … with region annotations (RefCOCO/… ) …. How does F-CAST compare to FLAIR under identical protocols?"
>
> In Figure 7, we showcase F-CAST's precise local-to-local alignment learned without direct supervision through the cross-attention maps between visual tokens and subcaptions. Indeed, these cross-attention maps can be quantitatively evaluated under two tasks: 1) Zero-shot Referring Image Segmentation and 2) Open-Vocabulary Semantic Segmentation. In the new experiments below, we find F-CAST shows consistently strong performance compared to the competitors, demonstrating its ability to 1) localize regions associated with a given text phrase and 2) identify which phrase corresponds to a given region.
>
> **Zero-shot Referring Image Segmentation**
>
> Given a target image and a referring text description, we compute the cross-attention map and apply Otsu thresholding [1] to obtain the final binary segmentation mask. We compare our method against models trained solely on image-text pairs without dense mask supervision like ours. Following the evaluation setting of SaG [2], we report mIoU on the RefCOCO, RefCOCO+, and GRef benchmarks below. F-CAST shows superior phrase localization capability over existing methods, including FLAIR.
>
> |Model|RefC|||RefC+|||GRef|
> |-|-:|-:|-:|-:|-:|-:|-:|
> ||val|testA|testB|val|testA|testB|val|
> |GroupViT|7.99|6.16|10.51|8.49|6.79|10.59|10.68|
> |MaskCLIP|11.52|11.85|12.06|11.87|12.01|12.57|12.74|
> |SaG [2]|21.80|19.00|24.96|22.20|19.86|24.85|25.89|
> |FLAIR|19.79|19.76|19.97|18.49|18.25|18.86|20.21|
> |F-CAST|**28.85**|**30.81**|**28.53**|**28.46**|**29.82**|**28.32**|**31.39**|
>
> **Open Vocabulary Semantic Segmentation**
>
> Given a target image and a list of candidate classes, we compute the cross-attention maps for each class separately, then assign each visual segment to the class with the highest response to obtain the final segmentation mask. Following FLAIR, we compare our method against vision-language models and evaluate mIoU on four segmentation benchmarks without background category setting. F-CAST consistently outperformed competitors across different data scales, including FLAIR.
>
> |Method|Data Size|VOC20|Cityscapes|Context59|COCO-Stuff|Average|
> |-|-:|-:|-:|-:|-:|-:|
> |CLIP|400M|41.8|5.5|9.2|4.4|12.8|
> |OpenCLIP|2B|47.2|5.1|9.0|5.0|13.9|
> |MetaCLIP|2.5B|35.4|5.0|8.1|4.3|11.0|
> |FLAIR|3M|60.9|20.6|23.8|13.1|26.3|
> |FLAIR|12M|69.7|20.1|22.9|15.4|28.3|
> |FLAIR|15M|66.7|16.5|17.4|13.6|24.7|
> |FLAIR|30M|73.0|13.6|18.6|13.3|25.8|
> |F-CAST|3M|70.5|23.4|25.3|13.6|33.2|
> |F-CAST|12M|74.7|23.2|26.9|14.7|34.9|
> |F-CAST|15M|74.1|**26.1**|**30.5**|13.6|36.1|
> |F-CAST|30M|**76.2**|24.7|28.6|**15.7**|**36.3**|
>
> &nbsp;
>
> [1] Otsu, N. A threshold selection method from gray-level histograms. IEEE Transactions on Systems, Man, and Cybernetics, 9(1), 62–66.
>
> [2] Kim et al. Shatter and gather: Learning referring image segmentation with text supervision. ICCV 2023.

---

> ### Author Response · Authors · 2025-11-21
>
> **[W3, Q2, Q4] Ablation study on part-level alignment loss**
>
> >  "The part-level pathway … is merely a training prior? A controlled experiment removes part-level loss at train time, …"
>
> > " … Conversely, if you remove L_part during training, holding compute constant, how much of Table 1 survives?"
>
> > " … What specific cases benefit from the part-level loss?"
>
> We would like to kindly note that this ablation study was already included in the appendix of our initial submission (Appx. E, Table 5). For your convenience, we include the same table from the appendix below.
>
> As shown, removing the part-level loss leads to a significant performance drop, demonstrating that fine-grained scene understanding requires detailed associations of local elements. We recognize the importance of this study, and we will move the table into our main paper in the revised manuscript.
>
> |Method|DCI I2T|DCI T2I|DOC I2T|DOC T2I|SV I2T|SV T2I|U1k I2T|U1k T2I|IIW I2T|IIW T2I|
> |-|-:|-:|-:|-:|-:|-:|-:|-:|-:|-:|
> |FLAIR (reproduced)|48.3|52.0|52.9|53.6|73.3|72.1|67.3|73.7|80.4|83.5|
> |**F-CAST (ours)**|**56.6**|**55.8**|**66.7**|**63.7**|**85.6**|**84.7**|**82.8**|**84.9**|**92.3**|**92.2**|
> |F-CAST w/o Part Align.|51.4|54.1|65.2|60.7|82.1|82.2|80.7|83.2|88.4|86.9|
>
> ---
>
> **[Q2, Q4] Inference with both whole- and part-level alignment (late fusion)**
>
> > "What happens if you retain the part-level branch at inference (e.g., late fusion ...) …"
>
> > " … Your inference uses only the global embedding. …"
>
> We adopt whole-level inference as the default scheme for image-text retrieval, following standard practice and for simplicity. Nonetheless, F-CAST can further incorporate part-level alignment scores on top of the whole-level alignment score for better retrieval performance. In particular, we compute the similarity between an image and a long caption using a weighted sum of the two scores, controlled by a hyperparameter $\alpha$.
>
> $$
> \text{score} =(1 - \alpha)  \cdot \text{whole score} + \alpha \cdot \text{mean(part score)}
> $$
>
>
> The tables below report the long-text image retrieval performance for different values of $\alpha$, where $\alpha=0.0$ corresponds to the original whole-level inference. As shown, combining the part-level score with the whole-level score yields better performance than using either score alone, suggesting that the two scores may partially compensate for each other. We observe that the part-level score tends to have a relative advantage in text-to-image retrieval, whereas the whole-level score tends to be more advantageous for image-to-text retrieval. A similar tendency is also observed when comparing FLAIR with its baseline LoTLIP (Tab. 1 in our main PDF), suggesting that text-conditioned attention-pooled image embeddings may introduce factors that negatively affect image-to-text retrieval performance. A deeper analysis of this behavior is an interesting direction that we leave for future work.
>
> |$\alpha$|DCI I2T|DCI T2I|DOCCI I2T|DOCCI T2I|SV1k I2T|SV1k T2I|SV10k I2T|SV10k T2I|Urban1k I2T|Urban1k T2I|IIW I2T|IIW T2I|
> |-|-|-|-|-|-|-|-|-|-|-|-|-|
> |1.0 (part only)|50.5|71.6|68.1|81.3|98.1|97.7|90.3|92.3|86.2|94.6|95.3|97.4|
> |0.7|63.2|**74.1**|77.1|**83.6**|**99.0**|98.4|94.3|94.3|92.2|**96.3**|97.4|**97.9**|
> |0.5|67.9|73.7|79.6|83.2|99.4|98.8|95.3|94.6|93.4|96.1|**97.6**|97.6|
> |0.3|**69.7**|72.4|80.4|82.0|**99.5**|**99.0**|**95.5**|**94.8**|**93.6**|95.4|**97.6**|97.4|
> |0.0 (whole only)|69.4|69.4|**80.6**|80.4|99.0|98.8|95.1|94.5|93.6|93.9|97.4|96.1|
>
> |$\alpha$   |I2T Avg|T2I Avg|Total Avg|
> |------------------|-------|-------|---------|
> |1.0 (part only)   |81.4   |89.2   |85.3     |
> |0.7               |87.2   |**90.8**|89.0    |
> |0.5               |88.9   |90.7   |**89.8**|
> |0.3               |**89.4**|90.2  |**89.8**|
> |0.0 (whole only)  |89.2   |88.9   |89.0     |

---

### Official Review · Reviewer_9jGL · 2025-10-31

**Soundness:** 3
**Presentation:** 3
**Contribution:** 2
**Rating:** 4
**Confidence:** 4

**Summary:**

The research addresses a critical limitation in current AI vision-language models: their inability to comprehend detailed, lengthy image descriptions containing specific details about objects, their attributes, and spatial relationships. The authors propose that effective understanding requires a hierarchical approach that mirrors human perception—simultaneously grasping both the overall scene and specific details.

Their solution, F-CAST, implements a three-stage hierarchical system that processes visual and textual information in parallel levels of granularity. The model progressively groups image patches into objects and then complete scenes while simultaneously analyzing individual sentences before combining them into full descriptions, ultimately aligning sentence-level details with corresponding image regions and matching complete captions with whole images. F-CAST is claimed to have achieved state-of-the-art results across six benchmarks and learned to associate specific phrases with image regions without explicit supervision.

**Strengths:**

Comprehension and matching long-text to images is an important and interesting problem. Also, and in general, the paper is well-written and technically sound.

**Weaknesses:**

As noted above, the challenge itself is not novel, nor is the solution strategy of joint learning. More detail on what is special about this specific manifestation of the problem and on the solution would have increase the novelty of this work.

**Questions:**

N/A

---

> ### Author Response · Authors · 2025-11-21
>
> Dear reviewer 9jGL,
>
> Thank you for your valuable feedback and comments. We appreciate your recognition of the importance of fine-grained vision-language understanding through long-text–to-image grounding, as well as your positive assessment of the clarity and technical soundness of our paper. In the response below, we clarify what is unique about our specific setting and provide additional details that highlight the distinctive aspects of our approach.
>
> ---
>
> **[W1] Detailed challenges and contributions**
>
> >  "The challenge itself is not novel, nor is the solution strategy of joint learning. More detail on what is special about this specific manifestation of the problem and on the solution would have increase the novelty of this work."
>
>
> **(1) Specific challenges we address**
>
> Vision-language models align images and captions as a whole, which makes them struggle with long, detailed descriptions. To achieve fine-grained vision language understanding, existing works aim to learn local associations across modality. However, they often rely on **additional dense annotations** and model cross-modal interactions in a  **flat manner** that ignores the hierarchical structure inherent in both vision and language. To address this limitation, we take a novel view that considers this challenge as a particular manifestation of jointly aligning visual and textual hierarchies.
>
> **(2) Novelty and contribution**
>
> Our core contribution is to introduce and realize a new paradigm for vision-language models (VLMs): the **joint alignment of visual hierarchy and textual hierarchy**. Prior works have either relied on flat textual encoders or used syntactic parse trees only on the text side. To the best of our knowledge, no prior work has explored representing **both** images and texts as hierarchical structures and **aligning the two hierarchies in a spatially oriented manner**.
>
> We show that this cross-modal hierarchical alignment becomes especially important when training with longer and more expressive captions. Our experiments demonstrate that this alignment substantially improves long-caption understanding (Table 1) and enables precise visual grounding where visual segments correspond to the textual subtrees (Figure 7).
>
> Importantly, our novelty does not lie in combining existing components to enhance performance on known tasks. Instead, the contribution is conceptual. We propose a **new architecture paradigm** based on joint hierarchical alignment. The use of FLAIR, CAST, or hierarchical transformers is simply one practical instantiation of this idea. We believe future work can adopt more advanced architectures while still following this paradigm.
>
> In summary, the novelty of our work is in 1) identifying the missing principle of cross-modal hierarchical alignment, 2) demonstrating its effectiveness for contemporary VLMs, and 3) providing a concrete implementation that validates this new perspective.
>
> ---
>
> We will more explicitly highlight the specific challenge we address and the novelty of our solution in the revised manuscript.
>
> We would be happy to clarify any remaining questions or concerns.
>
> Sincerely, Authors

---

### Official Review · Reviewer_AC3R · 2025-11-01

**Soundness:** 3
**Presentation:** 3
**Contribution:** 2
**Rating:** 4
**Confidence:** 4

**Summary:**

This paper proposes F-CAST, a hierarchical vision-language model that aligns fine-to-coarse visual structures with spatially oriented textual hierarchies for improved grounding in long-image caption understanding. By combining a CAST-based visual encoder and a two-stage hierarchical text transformer with a two-level alignment loss, F-CAST achieves state-of-the-art performance on six long-text image retrieval benchmarks without requiring region-sentence annotations.

**Strengths:**

1. The paper is well-written and easy to follow.
2. The proposed token reconstruction alignment and subcaption-aggregated patch alignment strategies are interesting and insightful.
3. Experimental results reflect the effectiveness of the proposed method to some extent.

**Weaknesses:**

I appreciate the contributions of this paper, but I have two main concerns—particularly the second one—that prevent me from giving a positive recommendation at this stage:

Although the F-CAST framework is interesting, its individual components appear to be borrowed directly from prior methods, essentially forming a combination of existing approaches.
F-CAST only reports quantitative results on long-text image retrieval benchmarks; its part-level capabilities are not sufficiently validated. I would expect the authors to align their evaluation with FG-CLIP or [1] and provide additional fine-grained results.

[1] UMG-CLIP: A Unified Multi-Granularity Vision Generalist for Open-World Understanding. ECCV, 2024.

**Questions:**

Please refer to the 'weakness' part.

---

> ### Author Response · Authors · 2025-11-21
>
> Dear reviewer AC3R,
>
> Thank you for your valuable feedback and comments. We appreciate your recognition of our clear presentation, the insightfulness of our hierarchical alignment strategy, and the effectiveness of F-CAST. We address your concerns in detail below.
>
> ---
>
> **[W1] Novelty and contribution**
>
> >  "… its individual components appear to be borrowed … forming a combination of existing approaches"
>
> Our core contribution is to introduce and realize a new paradigm for vision-language models (VLMs): the **joint alignment of visual hierarchy and textual hierarchy**. Prior works have either relied on flat textual encoders or used syntactic parse trees only on the text side. To the best of our knowledge, no prior work has explored representing **both** images and texts as hierarchical structures and **aligning the two hierarchies in a spatially oriented manner**.
>
> We show that this cross-modal hierarchical alignment becomes especially important when training with longer and more expressive captions. Our experiments demonstrate that this alignment substantially improves long-caption understanding (Table 1) and enables precise visual grounding where visual segments correspond to the textual subtrees (Figure 7).
>
> Importantly, our novelty does not lie in combining existing components to enhance performance on known tasks. Instead, the contribution is conceptual. We propose a **new architecture paradigm** based on joint hierarchical alignment. The use of FLAIR, CAST, or hierarchical transformers is simply one practical instantiation of this idea. We believe future work can adopt more advanced architectures while still following this paradigm.
>
> In summary, the novelty of our work is in 1) identifying the missing principle of cross-modal hierarchical alignment, 2) demonstrating its effectiveness for contemporary VLMs, and 3) providing a concrete implementation that validates this new perspective.

---

> ### Author Response · Authors · 2025-11-21
>
> **[W2] Quantitative validation of part-level capabilities**
>
> > "… its part-level capabilities are not sufficiently validated. …  I would expect the authors to align their evaluation with FG-CLIP …"
>
> Part-level alignment is a core component of F-CAST and is qualitatively studied in Section 4.3, where we show that F-CAST learns precise part-level grounding without any direct local supervision. Evaluating F-CAST using benchmarks from FG-CLIP or UMG-CLIP would be fundamentally unfair, as these methods rely on dense region-level annotations (bounding boxes paired with region descriptions) that are directly relevant to their task, whereas F-CAST uses no such labels. Nevertheless, to directly address the reviewer’s concerns, we evaluate our method on the Fine-Grained Open-Vocabulary Object Detection (FG-OVD) [1] benchmark, comparing only against methods that do not rely on dense labels during training. Models trained with dense annotations are treated as upper bounds.
>
> In addition, we further validate F-CAST’s part-level alignment ability through quantitative evaluations on standard benchmarks, including Zero-shot Referring Image Segmentation (ZRIS) and Open-Vocabulary Semantic Segmentation (OVSS), demonstrating that F-CAST's ability to 1) localize regions associated with a given phrase (ZRIS) and 2) identify which phrase corresponds to a given region (OVSS).
>
> **Fine-Grained Open-Vocabulary object Detection [1]**
>
> FG-OVD evaluates the ability to identify bounding box regions with fine-grained descriptions. Each bounding box has one positive description and ten hard-negative descriptions, and we report accuracy. The table highlights the superior fine-grained understanding of F-CAST compared to widely used Vision-Language Models such as CLIP, EVA-CLIP. Although models trained with long captions tend to lack regional understanding [1], F-CAST achieves a substantial performance margin over Long-CLIP, which we attribute to its visual grounding capability. For fair comparison, we separately report results for FG-CLIP and FineCLIP, as these methods were trained with bounding-box regions paired with region descriptions.
>
> |Method|hard|medium|easy|trivial|
> |-|-:|-:|-:|-:|
> |*Trained w/o dense annotations*|||||
> |CLIP|12.0|23.1|22.2|58.5|
> |EVA-CLIP|14.0|30.1|29.4|58.3|
> |Long-CLIP|9.2|18.4|16.2|51.8|
> |**F-CAST**|**16.8**|**32.8**|**38.6**|**62.5**|
> |*Trained w/ dense annotations*|||||
> |FineCLIP|26.8|49.8|50.4|71.9|
> |FG-CLIP|46.1|66.6|68.7|83.4|
>
> **Zero-shot Referring Image Segmentation**
>
> Given a target image and a referring text description, we compute the cross-attention map and apply Otsu thresholding [2] to obtain the final binary segmentation mask. We compare our method against models trained solely on image-text pairs without dense mask supervision like ours. Following the evaluation setting of SaG [3], we report mIoU on the RefCOCO, RefCOCO+, and GRef benchmarks below. F-CAST shows superior phrase localization capability over existing methods, including FLAIR.
>
> |Model|RefC|||RefC+|||GRef|
> |-|-:|-:|-:|-:|-:|-:|-:|
> ||val|testA|testB|val|testA|testB|val|
> |GroupViT|7.99|6.16|10.51|8.49|6.79|10.59|10.68|
> |MaskCLIP|11.52|11.85|12.06|11.87|12.01|12.57|12.74|
> |SaG [3]|21.80|19.00|24.96|22.20|19.86|24.85|25.89|
> |FLAIR|19.79|19.76|19.97|18.49|18.25|18.86|20.21|
> |F-CAST|**28.85**|**30.81**|**28.53**|**28.46**|**29.82**|**28.32**|**31.39**|
>
> **Open Vocabulary Semantic Segmentation**
>
> Given a target image and a list of candidate classes, we compute the cross-attention maps for each class separately, then assign each visual segment to the class with the highest response to obtain the final segmentation mask. Following FLAIR, we compare our method against vision-language models and evaluate mIoU on four segmentation benchmarks without background category setting. F-CAST consistently outperformed competitors across different data scales, including FLAIR.
>
> |Method|Data Size|VOC20|Cityscapes|Context59|COCO-Stuff|Average|
> |-|-:|-:|-:|-:|-:|-:|
> |CLIP|400M|41.8|5.5|9.2|4.4|12.8|
> |OpenCLIP|2B|47.2|5.1|9.0|5.0|13.9|
> |MetaCLIP|2.5B|35.4|5.0|8.1|4.3|11.0|
> |FLAIR|3M|60.9|20.6|23.8|13.1|26.3|
> |FLAIR|12M|69.7|20.1|22.9|15.4|28.3|
> |FLAIR|15M|66.7|16.5|17.4|13.6|24.7|
> |FLAIR|30M|73.0|13.6|18.6|13.3|25.8|
> |F-CAST|3M|70.5|23.4|25.3|13.6|33.2|
> |F-CAST|12M|74.7|23.2|26.9|14.7|34.9|
> |F-CAST|15M|74.1|**26.1**|**30.5**|13.6|36.1|
> |F-CAST|30M|**76.2**|24.7|28.6|**15.7**|**36.3**|
>
> ---
>
> We will add these new results to the revised manuscript if they are not already included.
>
> We would be happy to clarify any remaining questions or concerns.
>
> Sincerely, Authors
>
> &nbsp;
>
>
> [1] Bianchi et al. The devil is in the fine-grained details: Evaluating open-vocabulary object detectors for fine-grained understanding. CVPR 2024
>
> [2] Otsu, N. A threshold selection method from gray-level histograms. IEEE Transactions on Systems, Man, and Cybernetics, 9(1), 62–66.
>
> [3] Kim et al. Shatter and gather: Learning referring image segmentation with text supervision. ICCV 2023

---

### Official Review · Reviewer_yPGE · 2025-11-01

**Soundness:** 3
**Presentation:** 2
**Contribution:** 3
**Rating:** 6
**Confidence:** 3

**Summary:**

This paper proposes a hierarchical image-text representation learning framework, F-CAST, which enables vision-language models to learn hierarchically aligned information from images and long captions. The proposed method exceeds the SOTA methods on six long-text benchmarks.

**Strengths:**

- F-CAST learns fine-grained, visually grounded text understanding without supervision.
- The proposed methods exceeds the SOTA methods on six long-text benchmarks.

**Weaknesses:**

- F-CAST aims to improve the alignment between the visual hierarchy of images and the spatial hierarchy of long captions. However, the method is primarily evaluated on long text-image cross-modal retrieval tasks, which mainly assess global-to-global alignment in vision-language models. The hierarchical alignment capability is only illustrated through visualization rather than quantitative evaluation.
- The proposed method mainly focuses on local information at the sub-caption level. However, sub-captions often contain hierarchical information that may align with different visual regions. These multi-level relationships within sub-captions may not be fully captured by the proposed method. For example, in Figure 3, the first sub-caption also mentions “man” in addition to "horses."

**Questions:**

The paper mainly visualizes F-CAST’s ability to perform local alignment between sub-captions and image regions. What about the global alignment between the long text and the corresponding image?

---

> ### Author Response · Authors · 2025-11-21
>
> Dear reviewer yPGE,
>
> Thank you for your valuable feedback and comments. We appreciate your recognition of F-CAST's ability to learn fine-grained, grounded vision-language understanding through alignment between image and text hierarchies, as well as its SOTA performance across six challenging long-text–image retrieval benchmarks. We address your questions and concerns below.
>
> ---
>
> **[W1] Quantitative evaluation of local-to-local alignment**
>
> > "… The hierarchical alignment capability is only illustrated through visualization rather than quantitative evaluation."
>
> In Figure 7, we showcase F-CAST's precise local-to-local alignment learned without direct supervision through the cross-attention maps between visual tokens and subcaptions. Indeed, these cross-attention maps can be quantitatively evaluated under two tasks: 1) Zero-shot Referring Image Segmentation and 2) Open-Vocabulary Semantic Segmentation. In the new experiments below, we find F-CAST shows consistently strong performance compared to the competitors, demonstrating its ability to 1) localize regions associated with a given text phrase and 2) identify which phrase corresponds to a given region.
>
> **Zero-shot Referring Image Segmentation**
>
> Given a target image and a referring text description, we compute the cross-attention map and apply Otsu thresholding [1] to obtain the final binary segmentation mask. We compare our method against models trained solely on image-text pairs without dense mask supervision like ours. Following the evaluation setting of SaG [2], we report mIoU on the RefCOCO, RefCOCO+, and GRef benchmarks below. F-CAST shows superior phrase localization capability over existing methods, including FLAIR.
>
> |Model|RefCOCO|||RefCOCO+|||GRef|
> |-|-:|-:|-:|-:|-:|-:|-:|
> ||val|testA|testB|val|testA|testB|val|
> |GroupViT|7.99|6.16|10.51|8.49|6.79|10.59|10.68|
> |MaskCLIP|11.52|11.85|12.06|11.87|12.01|12.57|12.74|
> |SaG [2]|21.80|19.00|24.96|22.20|19.86|24.85|25.89|
> |FLAIR|19.79|19.76|19.97|18.49|18.25|18.86|20.21|
> |F-CAST|**28.85**|**30.81**|**28.53**|**28.46**|**29.82**|**28.32**|**31.39**|
>
> **Open Vocabulary Semantic Segmentation**
>
> Given a target image and a list of candidate classes, we compute the cross-attention maps for each class separately, then assign each visual segment to the class with the highest response to obtain the final segmentation mask. Following FLAIR, we compare our method against vision-language models and evaluate mIoU on four segmentation benchmarks without background category setting. F-CAST consistently outperformed competitors across different data scales, including FLAIR.
>
> |Method|Data Size|VOC20|Cityscapes|Context59|COCO-Stuff|Average|
> |-|-:|-:|-:|-:|-:|-:|
> |CLIP|400M|41.8|5.5|9.2|4.4|12.8|
> |OpenCLIP|2B|47.2|5.1|9.0|5.0|13.9|
> |MetaCLIP|2.5B|35.4|5.0|8.1|4.3|11.0|
> |FLAIR|3M|60.9|20.6|23.8|13.1|26.3|
> |FLAIR|12M|69.7|20.1|22.9|15.4|28.3|
> |FLAIR|15M|66.7|16.5|17.4|13.6|24.7|
> |FLAIR|30M|73.0|13.6|18.6|13.3|25.8|
> |F-CAST|3M|70.5|23.4|25.3|13.6|33.2|
> |F-CAST|12M|74.7|23.2|26.9|14.7|34.9|
> |F-CAST|15M|74.1|**26.1**|**30.5**|13.6|36.1|
> |F-CAST|30M|**76.2**|24.7|28.6|**15.7**|**36.3**|
>
> &nbsp;
>
> [1] Otsu, N. A threshold selection method from gray-level histograms. IEEE Transactions on Systems, Man, and Cybernetics, 9(1), 62–66.
>
> [2] Kim et al. Shatter and gather: Learning referring image segmentation with text supervision. ICCV 2023.

---

> ### Author Response · Authors · 2025-11-21
>
> **[W2] Multi-level hierarchy within sub-captions**
>
> > "… sub-captions often contain hierarchical information that may align with different visual regions. … These multi-level relationships within sub-captions may not be fully captured by the proposed method ... For example, in Figure 3, the first sub-caption also mentions “man” in addition to “horses”."
>
> Indeed, sub-captions may contain multiple semantic components that correspond to different visual regions. In Figure 10 of the revised manuscript, we newly include more detailed attention visualizations for the "man" and "horses" example. As shown,
> F-CAST effectively captures these multi-level hierarchical structures within sub-captions, despite not being explicitly trained for this. F-CAST not only localizes each object individually, but can also localize multiple objects simultaneously (e.g., both the "man" and the "horses"). Moreover, when we decompose a phrase (e.g., “a man wearing a polo shirt and holding a broom”) into smaller parts (e.g., “holding a broom”), F-CAST successfully grounds the corresponding part within the image.
>
> In this work, we mainly focus on the two-level spatial hierarchy that lies between the sub-caption and the full caption, but our design philosophy is not restricted by the number of hierarchies and can be naturally extended to deeper hierarchies. F-CAST is a first step toward even more fine-grained alignment beyond the sub-caption level (i.e., aligning forest, trees, and branches).
>
> ---
> **[Q1] Global alignment visualization**
>
> > "… What about the global alignment between the long text and the corresponding image?"
>
> In Figure 11 of the revised manuscript, we include new visualizations of global-level alignment between the full long caption and the image using CAM [3]. As shown, the global alignment tends to highlight the major regions of the image, in contrast to the more specific and concise regions produced by sub-caption alignment. This illustrates how sub-caption-level alignment naturally supports and contributes to the alignment of the whole caption.
>
> ---
> We will add these new results to the revised manuscript if they are not already included.
>
> We would be happy to clarify any remaining questions or concerns.
>
> Sincerely, Authors
>
> &nbsp;
>
> [3] Li et al. A Closer Look at the Explainability of Contrastive Language-Image Pre-training. Arxiv preprint 2023

---

### Author Response · Authors · 2025-12-04

Dear ACs,

Since interactions with reviewers are no longer available, we provide this brief summary of our rebuttal and emphasize that we have carefully addressed all concerns raised in the reviews.

**Reviewer-Acknowledged Strengths**:

All four reviewers acknowledged the soundness and effectiveness of our method, noting its contribution as a strong and scalable framework, with performance demonstrated across six benchmarks.

**Main Concerns and How We Resolved Them**:

1. Visual grounding evaluation: The primary concern involved the lack of certain quantitative results for visual grounding capabilities. In our rebuttal, we conducted extensive evaluations across three new benchmarks, closely following the reviewer’s suggested directions. The new results consistently demonstrate the grounding capabilities of our method and, we believe, convincingly address this concern.

2. Technical novelty: Some concerns appeared to stem from how our contribution was presented, influenced by our naming convention (F-CAST), which may have inadvertently drawn attention to pre-existing components (FLAIR and CAST) rather than our novel integration and learning paradigm. We have clarified these points comprehensively in our rebuttal.

We also responded in detail to all remaining points of concern raised in the reviews.

We would be grateful if you could pay special attention to the reviews and the points clarified in our rebuttal when making the final decision.

Yours sincerely, Authors of #23982

---

### Meta-Review · Area_Chair_vnEz · 2026-01-11

**Summary:**

This paper proposes F-CAST, a hierarchical vision–language pretraining framework that aligns fine-to-coarse visual structures with spatially oriented textual hierarchies to improve grounding in long-caption understanding. Reviewers agree the problem is important and the paper is clearly written. Initial concerns regarding lack of quantitative grounding evaluation, insufficient validation of part-level alignment and unclear articulation of novelty are partially addressed through additional experiments and clarifications in the author response. However, remaining concerns focus on whether the contribution is primarily a recombination of existing components and whether the analysis decisively establishes hierarchical grounding as a benefit. Given these important concerns about novelty and the strength of analysis for hierarchical alignment, the paper is not recommended for acceptance to ICLR. It is suggested that the authors fully incorporate the review suggestions towards a future submission.

**Reviewer Concerns:**

### Addressed concerns

* **yPGE:** The hierarchical alignment capability is only illustrated qualitatively rather than through quantitative grounding metrics. The author response adds extensive evaluations on zero-shot referring image segmentation and open-vocabulary semantic segmentation, demonstrating improved phrase-level localization.

* **yPGE:** Sub-captions may contain multiple semantic components that align to different visual regions. The author response provides new attention visualizations and examples showing that multiple objects within a single sub-caption (like “man” and “horses”) can be grounded simultaneously.

* **AC3R:** Lack of quantitative validation of part-level capabilities beyond retrieval. The author response adds evaluations on fine-grained open-vocabulary object detection, zero-shot referring image segmentation and semantic segmentation to address the request for part-level quantitative results.

### Unaddressed concerns

* **AC3R:** The method’s individual components appear to be borrowed from prior work, making the contribution primarily a combination of existing approaches rather than a new method. Although the author response reframes the contribution as a “new paradigm” of joint hierarchical alignment, it does not convincingly establish that the core technical ideas go beyond architectural recomposition.

* **9jGL:** The challenge itself is not novel and the joint-learning strategy is explored in prior work. While the author response elaborates how this setting is framed as hierarchical alignment, it does not fully resolve the reviewer’s concern that the problem formulation and solution strategy lack distinctiveness.

* **frwe:** Contribution is limited, with CAST, hierarchical text encoders, and sigmoid alignment losses closely following existing methods. The author response emphasizes the new architecture, but does not address the scope of new technical contributions.

* **AC3R:** Despite added experiments, the evaluation still centers primarily on retrieval. Grounding and hierarchy analysis are important for the paper, so beyond the added benchmarks, analysis should establish hierarchical alignment as the primary cause of performance gains.

**Reviewer Scores:**

* **yPGE:** Initial rating 6, concerns about lack of quantitative results for grounding are addressed, would likely maintain at 6.

* **AC3R:** Initial rating 4, major concerns about novelty and limited analysis of part-level alignment are partially resolved, would likely maintain at 4.

* **9jGL:** Initial rating 4, novelty concerns, would likely remain at 4.

* **frwe:** Initial rating 4, concerns persist on scope of contributions, would likely remain at 4.

---

### Decision · Program_Chairs · 2026-01-26

Reject